# Tiny Fish, Big Hope: Zebrafish Unlocking Secrets to Fight Parkinson’s Disease

**DOI:** 10.3390/biology14101397

**Published:** 2025-10-12

**Authors:** Manjunatha Bangeppagari, Akshatha Manjunath, Anusha Srinivasa, Sang Joon Lee

**Affiliations:** 1Zebrafish Drug Screening Center, Department of Cell Biology and Molecular Genetics, Sri Devaraj Urs Academy of Higher Education and Research (A Deemed to Be University), Tamaka, Kolar 563103, India; 2Center for Biofluid and Biomimic Research, Pohang University of Science and Technology (POSTECH), Pohang 37673, Republic of Korea

**Keywords:** zebrafish, Parkinson’s disease, neurotoxin, dopamine, neurons, paraquat, MPTP

## Abstract

Parkinson’s disease is a brain disorder that affects movement, causing symptoms such as shaking, stiffness, and slowed motion. Despite many years of research, there is still no cure, and current treatments only help to manage symptoms. To better understand how the disease develops and to find new treatments, scientists use model organisms—animals that can mimic aspects of human disease. One such animal is the zebrafish, a small, striped fish that shares many biological features with humans. In this article, we explore how zebrafish are being used to study Parkinson’s disease. These fish are beneficial because they are see-through when young, allowing scientists to observe changes in the brain and nervous system in real time. Zebrafish can also be genetically modified to carry changes seen in people with Parkinson’s, or exposed to chemicals that trigger similar symptoms. This makes them a powerful tool for testing potential drugs quickly and cost-effectively. Our review highlights how zebrafish are helping to uncover how the disease progresses and how they could lead to better, faster ways to find treatments. These insights may one day improve the lives of millions of people living with Parkinson’s disease.

## 1. Introduction

Parkinson’s disease (PD) is a complex neurodegenerative disorder characterised by the progressive loss of dopaminergic neurons in the substantia nigra region of the brain. This neuronal degeneration induces motor impairments, including tremors, rigidity, and bradykinesia [1,2]. Despite extensive research conducted over the past several decades, finding effective treatments for PD remains a significant challenge. This highlights the critical need for robust animal models to elucidate their underlying mechanisms and facilitate the development of targeted therapies [3,4]. Recently, zebrafish have emerged as a valuable model organism in PD research because of their genetic, physiological, and behavioral parallels with humans, enabling the modeling of PD-associated mutations in genes such as α-synuclein, Parkin, and LRRK2 [5,6]. Furthermore, zebrafish possess a dopaminergic system that closely resembles that of humans, allowing researchers to effectively study dopaminergic neuron function, degeneration, and responses to potential therapeutics within a biologically relevant context [7,8]. By leveraging the unique advantages offered by zebrafish models, scientists can gain deeper insights into PD pathophysiology and accelerate the development of effective treatments. This approach holds a significant promise for translating preclinical findings into clinical applications, ultimately benefiting patients affected by this debilitating disorder.

## 2. History of Zebrafish as a Model Organism

Zebrafish have emerged as a pivotal model organism in scientific research, especially in the fields of developmental biology and genetics [9,10]. The use of zebrafish as a model organism dates back to the 1960s. It was initially employed in embryological and genetic studies [11,12]. However, significant attention to zebrafish as a genetic model organism began in the 1980s, when Streisinger and his colleagues developed a technique for manipulating zebrafish embryos, laying strong groundwork for their extensive use in genetic studies [11,12].

The 1990s marked a critical milestone with the advent of transgenic methodologies [13]. Scientists have gained the ability to introduce foreign genes into zebrafish embryos, significantly enhancing the capacity to study gene function and regulation [13,14]. This breakthrough facilitated deeper insights into gene roles in various biological processes, including development, disease susceptibility, and their environmental responses [14]. In the 2010s, the introduction of advanced gene-editing technologies, notably CRISPR/Cas9, greatly expanded the utility of zebrafish [15]. These precise genetic tools allowed researchers to generate zebrafish models carrying targeted gene mutations, modifications that enabled researchers to create zebrafish models with specific gene mutations or alterations, which have become essential for understanding disease, indispensable for examining the functions of individual genes in disease development, and for identifying potential therapeutic targets.

## 3. Applications in Development and Disease Research

Zebrafish models have been extensively employed in developmental biology research, providing critical insights into fundamental biological processes, such as organ formation, tissue regeneration, and embryonic patterning [16]. Their genetic versatility, optical transparency, and relatively short generation time make them invaluable for unravelling the complexities of vertebrate development [11,14]. Researchers can directly observe developmental processes in real-time and conduct large-scale genetic screens efficiently. Beyond development, zebrafish have significantly contributed to disease modelling, drug discovery, and toxicology [16]. They are used to model various human diseases, including cardiovascular disorders, cancer, neurodegenerative diseases, and developmental abnormalities [17]. Zebrafish embryos are highly sensitive to environmental toxins and pharmaceutical compounds, making them particularly useful for assessing drug efficacy and safety profiles [17,18]. These models have facilitated the identification of potential therapeutic agents and have advanced our understanding of disease mechanisms, substantially supporting medical research and pharmacology.

## 4. Advances in Neurobiology and Imaging

The optical transparency of zebrafish embryos and larvae has enabled sophisticated imaging approaches in neuroscience. In vivo calcium imaging has been widely used to study neuronal function and development, allowing observation of brain-wide activity and specific neuronal populations with synthetic and genetically encoded calcium indicators such as GCaM [19,20]. These methods provide crucial insights into the functions of disease-related genes.

Furthermore, zebrafish are highly amenable to optogenetics, which allows precise manipulation and observation of neural circuits. This has been applied in studies of neurodegenerative diseases such as ALS, where zebrafish models help elucidate motor neuron degeneration and potential therapeutic strategies [21]. Combining optogenetics with calcium imaging enables investigation of causal links between neural activity and behavior, as shown in recent studies involving brain-wide activity and targeted stimulation in freely swimming zebrafish [22].

The development of specialized zebrafish lines has further expanded research possibilities. For instance, the crystal mutant, which lacks pigmentation in both the body and eyes, provides enhanced optical access for imaging neural activity and behavior, overcoming the limitations of traditional pigmentation mutants [23].

## 5. Zebrafish as a Model Organism to Study Parkinson’s Disease

Zebrafish have become an important system for modeling Parkinson’s disease (PD) because of their genetic similarity to humans, transparent embryos that enable real-time imaging, and suitability for both toxin-based and genetic approaches [24,25,26].

Neurotoxins such as MPTP (1-methyl-4-phenyl-1,2,3,6-tetrahydropyridine), rotenone, and paraquat are commonly used to induce PD-like pathology in zebrafish [24]. MPTP is metabolized in glial cells to MPP^+^ (1-methyl-4-phenylpyridinium), which enters dopaminergic neurons via the dopamine transporter. Once inside, MPP^+^ disrupts mitochondrial function, promotes oxidative stress through interactions with redox-active metals such as iron, and leads to selective dopaminergic cell death in the substantia nigra [24,27,28,29,30]. Rotenone and paraquat similarly induce oxidative stress and mitochondrial dysfunction, resulting in dopaminergic degeneration [31,32]. In addition, toxin exposure in zebrafish activates inflammatory pathways including HMGB1, TLR4, and NFκB, processes that parallel inflammatory mechanisms implicated in PD progression [33].

In parallel with toxin-based models, transgenic zebrafish lines expressing human α-synuclein have been developed to study protein aggregation. The misfolding and accumulation of α-synuclein, a hallmark of PD, leads to Lewy body formation and dopaminergic neurotoxicity [6,7,34,35,36]. These models enable detailed investigation of α-synuclein toxicity and its cellular consequences.

Zebrafish have also been used to examine the roles of other PD-associated genes. Knockdown of pink1 and park2 disrupts mitophagy and dopaminergic neuron survival, reflecting mitochondrial dysfunction seen in human PD [33,37]. Similarly, dj1 knockdown increases vulnerability to oxidative stress, underscoring its neuroprotective role [37]. Together, these genetic findings align with toxin models, showing how variations in mitochondrial and dopamine transporter function can influence susceptibility to MPTP-induced neurodegeneration [38].

The development of modern genetic tools has expanded the use of zebrafish in PD research. Techniques such as CRISPR-Cas9, morpholino antisense oligonucleotides, and TILLING allow precise modeling of PD-associated mutations in genes such as parkin, PINK1, DJ-1, and LRRK2 [39]. These genetic approaches complement toxin-based models and provide powerful systems to explore the molecular mechanisms underlying PD.

Together, neurotoxin-induced and genetic zebrafish models provide complementary insights into PD pathogenesis. While zebrafish cannot fully replicate the chronic and progressive nature of human PD, their combined use has greatly enhanced our understanding of mitochondrial dysfunction, protein aggregation, and neuroinflammation in PD. Importantly, these models support the identification of therapeutic targets and the development of novel treatment strategies [40,41] (Figure 1 and Table 1).

## 6. Inducing Parkinson’s Symptoms in Zebrafish Using Parquat

The induction of PD-like symptoms in zebrafish using paraquat (PQ) relies on its ability to generate reactive oxygen species (ROS), leading to oxidative stress within cells [31,42]. PQ exposure elevates ROS levels, resulting in oxidative damage to lipids, proteins, and DNA [43,44]. These effects particularly compromise dopaminergic neurons, which are highly vulnerable to oxidative injury and are central to motor control [7,15,45,46,47,48]. In addition to oxidative stress, PQ induces neuroinflammation by activating microglia, the brain’s resident immune cells, which in turn release pro-inflammatory cytokines and trigger signalling cascades that exacerbate neuronal damage [45,48,49]. Mitochondrial dysfunction is another key component of PQ toxicity, as impaired energy production increases the oxidative burden and accelerates dopaminergic neuronal degeneration [45,46,50,51]. The convergence of oxidative stress, neuroinflammation, and mitochondrial dysfunction thus explains the pathological mechanisms underlying PQ-induced PD in zebrafish [40].

Beyond these molecular mechanisms, PQ exposure produces measurable Parkinsonian phenotypes. Zebrafish treated with PQ exhibit significant locomotor deficits, including reduced swimming activity, impaired exploratory behaviour, and increased aggression, which parallel key motor symptoms of PD in humans [52,53]. PQ has also been shown to influence α-synuclein homeostasis by shifting it toward monomeric and aggregation-prone forms, providing a platform to study synucleinopathy-related mechanisms in vivo [54]. These quantifiable outcomes can be captured with straightforward protocols and low-cost video tracking software, such as ToxTrac (version v2.61), making PQ models accessible and cost-effective for neurobehavioral studies [52].

Despite these advantages, PQ models have important limitations. The toxin’s systemic cytotoxicity extends beyond the dopaminergic system, reducing specificity compared with other neurotoxins [18]. Strain-dependent variability in response and differences between larval and adult zebrafish also complicate reproducibility and translational relevance [55] (Figure 2).

Following the behavioral assessment methods summarized in Table 2, several studies have demonstrated how these assays translate into measurable phenotypes in zebrafish models of Parkinson’s disease (PD).

### 6.1. MPTP Exposure

Administration of 1-methyl-4-phenyl-1,2,3,6-tetrahydropyridine (MPTP) results in marked motor impairments, including reduced swimming velocity and increased freezing behavior, which mirror the locomotor deficits characteristic of PD [56,63]. At the molecular level, MPTP exposure alters the expression of PD-associated genes and proteins, with downregulation of NEFL and MUNC13-1 implicated in disrupted neurological pathways [63].

### 6.2. Rotenone Exposure

Developmental exposure to rotenone in zebrafish embryos induces pathological features such as muscle atrophy and impaired motor performance. These changes reflect both motor and non-motor dimensions of PD, highlighting the contribution of environmental toxins to disease pathology [64].

### 6.3. Genetic Studies

Zebrafish models have also been employed to investigate genetic contributors to PD. These studies provide critical insights into the interplay between genetic predisposition and environmental triggers, shedding light on disease etiology and offering new avenues for therapeutic intervention [65].

### 6.4. Paraquat-Induced Symptoms

Paraquat exposure induces a spectrum of PD-like phenotypes. Motor impairments include bradykinesia, tremor, postural instability, and rigidity, while cognitive decline manifests through deficits in memory and executive function, resembling features shared with Alzheimer’s disease [66]. Non-motor symptoms such as mood disturbances, sleep irregularities, and autonomic dysfunction further align with the complex clinical presentation of PD [66].

### 6.5. Shared Pathological Mechanisms

Across toxin-induced and genetic models, zebrafish recapitulate several fundamental mechanisms of neurodegeneration. Oxidative stress driven by reactive oxygen species is a major driver of neuronal damage [67]. Chronic neuroinflammation contributes to progressive dopaminergic loss [68], while mitochondrial dysfunction remains central to impaired energy metabolism and apoptosis [69]. These conserved processes reinforce the translational relevance of zebrafish PD models.

### 6.6. Behavioral Assessments in Practice

Consistent with the assays detailed in Table 2, zebrafish exposed to PD toxins exhibit robust motor deficits, including reduced swimming speed and distance [57,62]. Non-motor phenotypes are equally prominent: light–dark box testing reveals anxiety-like behavior, and maze-based tasks demonstrate learning impairments and poor decision-making following rotenone exposure [59].

### 6.7. Neurochemical Analyses

These behavioral outcomes are supported by neurochemical data. Fast-scan cyclic voltammetry has shown significant reductions in dopamine release in toxin-exposed zebrafish, directly linking impaired motor function to dopaminergic deficits [59]. Gene expression analyses further indicate that while some dopamine metabolism genes remain unchanged, overall neurochemical profiles are significantly altered, reflecting the broader impact of neurotoxin exposure on neural circuits [57].

AD patients often experience significant memory loss and cognitive impairment, while PD patients may develop cognitive deficits, especially in later stages [70,71]. PD is characterized by tremors, rigidity, and bradykinesia. In contrast, AD can lead to motor impairments as the disease progresses [72]. Neuropsychiatric Symptoms: Both diseases can manifest depression, anxiety, and behavioral changes, complicating the clinical picture [70,71]. Oxidative Stress: Increased oxidative stress is common in both diseases, with dopamine oxidation in PD and amyloid-beta accumulation in AD contributing to neuronal damage [73,74]. Mitochondrial Dysfunction: Impaired mitochondrial function disrupts energy metabolism and promotes cell death in both conditions [69,73]. Neuroinflammation: Chronic neuroinflammation, driven by pro-inflammatory cytokines and DAMPs, is observed in both diseases [73,74].

## 7. Advantages of Zebrafish as a PD Model

Zebrafish have emerged as a powerful model organism in PD research, offering several advantages that make them highly suitable for genetic contributions [75,76,77]. Their genetic similarity to humans, especially in key PD-related genes such as α-synuclein, LRRK2, and Parkin, allows researchers to model mutations associated with PD pathogenesis [77,78]. This genetic tractability is essential for how specific mutations contribute to disease progression and for identifying molecular targets for therapeutic intervention [7,79,80,81,82].

One of the most distinctive advantages of zebrafish is the transparency of their embryos, which allows real-time visualisation of neuronal development, dopaminergic circuit formation, and drug effects [83,84]. This feature facilitates advanced imaging techniques, such as confocal and live-cell microscopy, enabling detailed observation of PD-related changes at both the cellular and subcellular levels [85,86]. Researchers can monitor disease progression over time and evaluate treatment responses in a living system.

Additionally, zebrafish are ideally suited for high-throughput drug screening due to their rapid development, large clutch size, and low maintenance cost [85,86,87]. These attributes make it feasible to test hundreds of compounds simultaneously, expediting the discovery of potential therapeutics and enabling early evaluation of efficacy and toxicity [43]. Their amenability to precise genetic manipulation further supports the functional validation of PD-related genes and enhances the development of targeted treatment strategies [6,85].

Overall, zebrafish combine genetic manipulability, conserved molecular pathways, reproducibility, and scalable experimentation. These qualities make them a robust model for exploring PD pathophysiology, testing therapeutic compounds, and contributing to translational advances in neurodegenerative disease research [16,88].

### Success Stories

Modeling Neurodegenerative Diseases: Zebrafish have been successfully used to model PD, showcasing key features such as protein aggregation and neuronal degeneration, which are critical for understanding disease mechanisms [89].

The transparent embryos of zebrafish facilitate real-time imaging and allow for the screening of numerous compounds simultaneously, leading to the identification of potential therapeutic agents for PD [90]. Compounds identified through zebrafish models have progressed to preclinical studies, demonstrating their relevance in developing effective treatments for neurodegenerative conditions [16] (Table 3).

## 8. Limitations and Considerations

Zebrafish indeed exhibit simpler behaviours compared to mammals, posing a challenge in assessing complex motor symptoms and cognitive deficits, typical indications of PD patients [93,94]. While zebrafish show basic locomotor behaviours and stimulus responses, they lack the advanced motor control and cognitive functions observed in mammals [95,96]. This limitation can be complemented by utilising supplementary models, such as rodents or non-human primates, which display more sophisticated behaviours resembling those observed in PD patients [98]. These alternative models can be utilised to investigate higher-order brain functions, including cognitive impairments associated with PD symptoms.

Another limitation in employing zebrafish models for PD research is their lack of a mammalian-like nigrostriatal pathway [45,85]. The nigrostriatal pathway is pivotal for dopamine signalling and plays a central role in PD pathogenesis due to the degeneration of dopaminergic neurons [99,100]. The administration of targeted toxin can help address this limitation by selectively inducing neuronal degeneration in specific brain regions, replicating the aspects of dopaminergic neuron loss observed in PD [7]. Furthermore, advancements in genetic engineering allow for the creation of zebrafish models with altered neuronal pathways, enabling more precise and targeted investigations on PD-related mechanisms [7,101].

Zebrafish models also possess an immature immune system compared to mammals, potentially affecting neuroinflammatory responses and immune-mediated aspects of PD pathogenesis [102]. Neuroinflammation is increasingly recognised as a critical contributor to neurodegeneration in PD [103]. Therefore, the incorporation of immune modulation strategies into zebrafish PD models is required to accurately reflect immune-mediated processes relevant to PD pathology [102,104].

Despite these limitations, zebrafish remain a valuable model for PD research, providing insights into fundamental molecular mechanisms and the search for potential therapeutic targets [84,85]. Nevertheless, it is strongly required to figure out these constraints and integrate the findings from zebrafish studies with the results obtained from other animal models and clinical data to achieve a comprehensive understanding of PD complexity and to develop effective therapeutic strategies. Ethical considerations on the use of zebrafish as an animal model must also be carefully addressed to ensure responsible and humane scientific practices [105].

## 9. Future Directions

Zebrafish models hold significant promise for advancing therapeutic development in PD research [6,7,85]. They provide a versatile platform for screening drug candidates and identifying novel molecular targets [106]. PD-like symptoms can be induced in zebrafish through neurotoxin exposure or genetic manipulation, allowing researchers to evaluate the efficacy of potential treatments rapidly and cost-effectively [45,94]. Their fast development, low maintenance, and compatibility with high-throughput assays make zebrafish especially valuable for early-stage drug discovery [103,107]. Additionally, their optical transparency enables real-time visualisation of neurodegenerative changes, offering insights into disease mechanisms and pharmacological responses [7,9,99].

Recent advances have expanded the potential of zebrafish in supporting personalised medicine approaches. Their genetic tractability allows for the creation of models that mirror specific patient mutations, facilitating individualized assessments of treatment efficacy [16,76]. These models also enable exploration of gene–environment interactions, which is essential for understanding the heterogeneous nature of PD and tailoring therapies accordingly [16]. Personalised zebrafish models represent a powerful strategy for improving patient-specific outcomes [76,88].

Moving forward, several specific directions could improve the translation potential to human biology. First, zebrafish genetic and chemical PD models should be optimised to capture human-relevant disease processes. Knockdowns and mutations in PARKIN, PINK1, DJ-1, and SNCA have already shown strong parallels to human pathology, while the use of MPTP in adult zebrafish has been highlighted as a means to better replicate age-related PD [46,55]. Such refinements are important for modelling the progressive and late-onset features of PD. Moreover, novel zebrafish studies have revealed pathological mechanisms, including cytosolic leakage of mitochondrial DNA and pathogenic phosphorylation of α-synuclein, which provide new translational targets for therapy [92].

Second, advances in technological tools will strengthen the relevance of zebrafish models. Optogenetics can be applied to manipulate specific neuronal circuits and examine their role in motor and cognitive dysfunction in PD [108]. Automated behavioural platforms, such as the Z-LaP Tracker, enable precise quantification of motor and cognitive impairments and support large-scale drug screening [109]. In addition, genome-editing techniques such as CRISPR/Cas9 allow the development of zebrafish lines with patient-specific mutations, improving the fidelity of zebrafish as a translational tool for human PD [110].

Third, future research should prioritize standardization and collaboration. The establishment of consistent behavioural assays and imaging protocols across laboratories will improve reproducibility and facilitate cross-study comparisons [110]. Interdisciplinary approaches, including combining zebrafish studies with mammalian models, may help overcome species-specific limitations and strengthen the translational bridge to human biology.

Finally, zebrafish models should be strategically integrated into drug discovery pipelines. Their high-throughput screening capabilities have already proven effective in identifying candidate compounds [89,108]. Applying these pipelines to therapies targeting mitochondrial dysfunction, synuclein aggregation, or neuroinflammatory pathways could accelerate the discovery of treatments with greater clinical relevance.

## 10. Conclusions

Zebrafish have become a cornerstone in PD research, offering unique advantages for studying the disease’s underlying mechanisms and for developing new therapies. Their amenability to genetic manipulation enables scientists to model specific gene mutations associated with PD, while their transparent embryos facilitate real-time visualization of dopaminergic neuron development and degeneration. Owing to the conservation of key molecular pathways between zebrafish and humans, findings from zebrafish models hold significant translational value. These models have contributed to the identification of disease-related genes and the discovery of candidate compounds that may protect or restore dopaminergic function. Additionally, the species’ rapid development and high fecundity support efficient drug testing pipelines.

Despite these strengths, zebrafish do present limitations. Their nervous system is structurally simpler than that of mammals, which can make modelling complex motor symptoms and higher-order cognitive dysfunctions more challenging. Moreover, their lack of a mammalian-like nigrostriatal pathway and an immature immune system may limit their use in modelling certain PD features. Nonetheless, when used in combination with rodent or primate models, zebrafish can provide complementary insights. A multi-model strategy that integrates zebrafish findings with data from higher organisms will enhance our overall understanding of PD and accelerate the development of effective, targeted treatments for patients.

## Figures and Tables

**Figure 1 biology-14-01397-f001:**
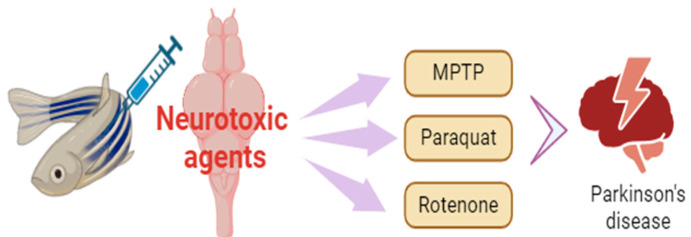
Zebrafish as a Parkinson’s disease model.

**Figure 2 biology-14-01397-f002:**
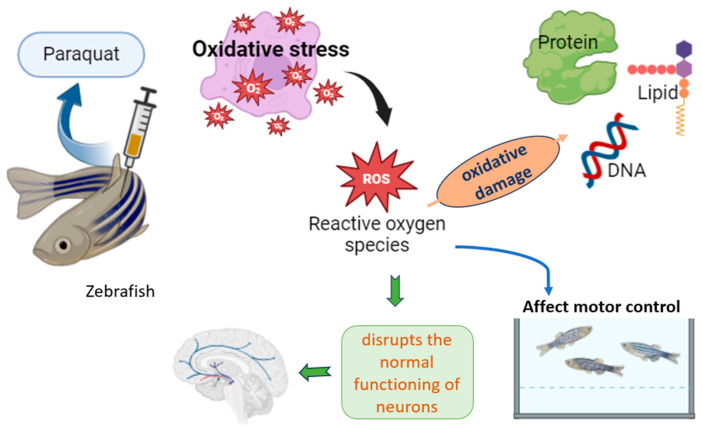
Drug-inducible mechanisms of inducing Parkinson’s symptoms in a zebrafish model.

**Table 1 biology-14-01397-t001:** Classification of zebrafish PD models with stages and applications.

Model Category	Typical Stage Used	Application
Neurotoxin (MPTP, Rotenone, Paraquat, 6-OHDA)	Larvae for throughput; Adults for chronic/behavioral	Larvae: imaging & high throughput; Adults: complex motor assays
Genetic (SNCA, PINK1, Parkin, LRRK2, DJ-1)	Embryo/larva → adult (depending on phenotype)	Early developmental effects in larvae; adult lines for progressive phenotypes
Environmental (Mn, Pb, pesticide mixtures)	Larvae and adults (dose/time dependent)	Models cumulative, low-dose or chronic exposures; behavioural impact in adults

**Table 2 biology-14-01397-t002:** Behavioral assays in zebrafish PD models capture both motor and non-motor symptoms relevant to human disease.

Category	Method	Protocol Overview	Key Parameters	Typical Equipment	Validation/Notes
Motor	Open-field locomotor assay (adult)	Single fish in arena, recorded 0–96 h post-toxin (MPTP, rotenone)	Distance, velocity, immobility, turn angle, meander	Video camera, arena, tracking software	Validated against DA depletion and TH staining [46,55,56,57,58]
Motor	Automated larval swimming tracking	Larvae exposed to MPP+ or rotenone; monitored under light/dark cycles	Swim distance, bout counts, thigmotaxis, transitions	Multi-well plates, automated imaging/tracking	Dose–response and drug rescue shown [46,56,57,58]
Motor	Maze & reward latency tests	Fish trained to reach reward after toxin exposure	Latency, errors, path efficiency, learning curve	Custom maze, video tracking	Cognitive impairments linked to DA release deficits [55,59,60]
Motor	Kinematic analysis & acoustic startle	High-speed capture of escape/startle; acoustic pulses	Turn duration, angular velocity, startle latency/habituation	High-speed camera, acoustic stimulator	Sensitive to subtle sensorimotor + deficits [46,56,61].
Motor	Electrical stimulation (microfluidic)	6-OHDA larvae; electrical pulses in lab-on-chip	Evoked locomotor amplitude, response frequency	Microfluidic chip, electrodes, video	Validated with Panx1 mutants & TH analysis [58,61]
Non-motor	Light–dark preference	Fish explore divided tank	Time in zones, transitions, latency	Light/dark box, tracker	Anxiety-like phenotypes observed [55,57,60,62]
Non-motor	Thigmotaxis	Open field with center/periphery zones	Wall-following, time in center vs. periphery	Arena, tracker	Reliable anxiety measure in PD models [46,58,62]
Non-motor	Sleep & circadian monitoring	24 h continuous recording	Sleep duration, latency, fragmentation, circadian phase	Infrared cameras, automated software	Melatonin rescue of sleep deficits shown [55,56,57,62]
Non-motor	Social interaction/shoaling	Paired/group assays or mirror tests	Interaction time, aggression, shoaling	Dual chamber, video	Rotenone reduces sociality, increases aggression [60,62]
Non-motor	Olfactory response testing	Odor choice/gradient assays	Latency, preference index, discrimination	Olfactometer, airflow, video	Limited but reported in MPTP/rotenone [46,56]
Non-motor	Cognitive (conditioning) assays	Classical/operant learning, memory retention	Acquisition, retention, reversal learning	Conditioning chambers, stimulus system	MPTP/rotenone impair memory; rescued by drugs [55,59,60]

**Table 3 biology-14-01397-t003:** Zebrafish models of Parkinson’s disease: approaches, phenotypes, comparative advantages over rodent systems, and limitations.

Modeling Approach	Key Phenotypes in Zebrafish	Comparative Advantages vs. Rodent Systems	Limitations	References
Neurotoxin: MPTP, 6-OHDA	Dopaminergic neuron loss; reduced locomotion; erratic swimming; altered gene/protein expression in neurological pathways	Rapid induction of PD-like symptoms; transparent larvae allow real-time imaging; cost-effective and scalable for drug screening	May not fully capture chronic or late-onset features of PD	[24,27,28,31,32,37]
Neurotoxin: Rotenone, Paraquat	Oxidative stress; mitochondrial dysfunction; progressive dopaminergic neurodegeneration; motor impairments	Mimics environmental toxin exposure in humans; models oxidative stress mechanisms effectively	Toxicity profiles differ from mammals; long-term exposure studies are limited	[31,32,37]
α-Synuclein transgenic lines	Protein aggregation; Lewy body-like inclusions; dopaminergic cell loss	Directly models hallmark human PD pathology; optical transparency allows tracking of aggregation in vivo	Zebrafish lack endogenous α-synuclein homolog, requiring transgenic approaches	[6,7,34,35,36,91,92]
PINK1/Parkin knockdown or mutants	Defective mitophagy; dopaminergic neuron vulnerability; motor dysfunction	Conserved mitochondrial pathways; faster assessment of mitophagy compared to rodents	Early-onset PD mutations may not model late-onset disease well	[33,37,46]
DJ-1 knockdown	Increased susceptibility to oxidative stress; dopaminergic cell loss	Mechanistic insight into oxidative stress pathways in PD	Partial phenotype compared to human PD	[37,91,92]
LRRK2 mutant lines	Synaptic dysfunction, altered vesicle trafficking, impaired autophagy	Models familial PD mutations; allows rapid in vivo functional assays	Some phenotypes are less pronounced than in mammalian models	[46]
Other genetic knockdowns (dj1, pink1, prkn)	Familial PD-like phenotypes; altered dopaminergic pathways	Stronger phenotypic expression than rodents in some cases; genetic tractability	Require validation against human disease heterogeneity	[37,91]
Drug screening/high-throughput assays	Behavioural rescue; reduced aggregation; restored mitochondrial function	Transparent embryos allow in vivo pharmacology; a scalable, cost-effective alternative to rodent models	Differences in metabolism and lifespan limit direct translation	[93,94,95,96]
Comparative advantages	Real-time imaging; rapid development; high-throughput screening	Cost-effective, ethically favourable, and genetically tractable	Short lifespan, lack of some human-specific proteins (e.g., α-synuclein)	[7,46,89,97]

## Data Availability

No new experimental data were created.

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
