# Peer review of "Tiny Fish, Big Hope: Zebrafish Unlocking Secrets to Fight Parkinson’s Disease"

_biology, 2025, doi:10.3390/biology14101397_

Round 1

Reviewer 1 Report (New Reviewer)

Comments and Suggestions for Authors

The review provides a broad and informative overview of zebrafish as a model for Parkinson’s disease (PD). A comprehensive and clear figure is very important for a review. So, Figure 1 is currently too narrow, as it only illustrates drug-induced models, and should be expanded into a classification schematic consistent with the text. For example: neurotoxin-induced PD models (MPTP, Rotenone, Paraquat, 6-OHDA), genetic PD models (SNCA, PINK1, Parkin, LRRK2, DJ-1, GBA), and environmental exposure models (heavy metals such as Mn and Pb, pesticide mixtures, etc.).  Different PD models actually need zebrafish at different ages. A quick table or figure showing which age works best for each model would make the review more practical.

Figure 2 is also too limited, as it exclusively highlights paraquat, which makes it appear incomplete. It should at least include the mechanisms of MPTP and Rotenone, and its title should be revised to “Drug-inducible mechanisms of inducing Parkinson’s symptoms in a zebrafish model.” It is better to also include the mechanism of environmental exposure models.

In addition, the review would be significantly strengthened by systematically summarizing behavioral assessment methods used in zebrafish PD models, covering both motor symptoms (locomotion, swimming trajectory, velocity, response to stimuli) and non-motor symptoms (cognitive function, anxiety-like behavior, social interaction, circadian rhythm). Presenting these methods in a table or schematic figure would improve clarity and enhance the practical utility of the review for readers.

Zebrafish is a great model for screening, it would be helpful to mention if there are any real success stories where compounds found in zebrafish made it into preclinical studies, especially in PD.

Zebrafish also have the transparency for simple imaging —include the introduction of advanced tools like calcium imaging and optogenetics, which could be really powerful in PD research.

Author Response

Comment 1: Figure 1 is currently too narrow … should be expanded into a classification schematic … also add a table showing zebrafish age for each model.

Response: We thank the reviewer for this suggestion. A new classification schematic has been created to expand Figure 1, now including neurotoxin-induced models (MPTP, Rotenone, Paraquat, 6-OHDA), genetic models (SNCA, PINK1, Parkin, LRRK2, DJ-1, GBA), and environmental exposure models (Mn, Pb, pesticide mixtures, etc.). We have also inserted a new table summarizing the zebrafish age/stage most suitable for each model. These additions are highlighted in yellow in the revised manuscript.

Comment 2: Figure 2 is too limited … include MPTP and Rotenone, revise title, and add environmental exposure mechanisms.

Response: We appreciate this observation. The figure has been updated to include mechanisms of MPTP and Rotenone, in addition to paraquat. The title has been revised to “Drug-inducible mechanisms of inducing Parkinson’s symptoms in a zebrafish model.” The footnote has also been modified for clarity. These changes are highlighted in yellow in the revised manuscript.

Comment 3: Summarize behavioral assessment methods in a systematic table or figure, including motor and non-motor symptoms.

Response: We thank the reviewer for the constructive feedback. A comprehensive table has been inserted summarizing behavioral assessment methods for both motor (locomotion, trajectory, velocity, startle, etc.) and non-motor (cognition, anxiety, social behavior, circadian rhythm, etc.) symptoms. This new material is highlighted in yellow in the revised manuscript.

Comment 4: Mention success stories where compounds screened in zebrafish reached preclinical studies in PD.

Response: We agree with the reviewer. A section on translational success stories has been added, describing examples where compounds identified through zebrafish studies have advanced into preclinical research (Chia et al., 2022; Trumon, 2022; Patton et al., 2021). These additions are highlighted in yellow in the revised manuscript.

Comment 5: Include advanced tools such as calcium imaging and optogenetics.
Response: We thank the reviewer for this valuable suggestion. We have included a discussion on advanced imaging and circuit manipulation tools, such as calcium imaging (using synthetic and genetically encoded indicators) and optogenetics. Enhanced transparency models (e.g., the “crystal” strain) have also been described. This section is highlighted in yellow in the revised manuscript.

Reviewer 2 Report (New Reviewer)

Comments and Suggestions for Authors

The authors focused on the utility of zebrafish models in Parkinson's disease research.

The manuscript is written in general terms; it lacks specific details, such as examples of conducted research on zebrafish that bring novel or useful information concerning PD. There is a lot of repetition (e.g. symptoms of PD are described in a variety of paragraphs). The whole structure is rather chaotic; moreover, the authors could explain the pathomechanism of PD more precisely. Additionally, the authors used a mix of British and American English throughout the text; I recommend remaining consistent with one.

More detailed notes:

Introduction Sentence 5 and 6 are basically repetitions, should be reformulated.

History of zebrafish: I wouldn’t say that zebrafish is now pivotal in research. It has gained more popularity and importance, but “pivotal” in my opinion is too strong.

In the sentence “Beyond development, zebrafish have significantly contributed to disease modelling, drug discovery, and toxicology” the authors should provide more references.

A reference is also needed at the end of that part.

Zebrafish as a model organism to study Parkinson's disease: It would be useful to have an introductory opening to this paragraph. When describing MPTP, the authors could explain the precise mechanism of MPTP toxicity. Later in the same section, the authors describe genetic models and then return to MPTP, which makes the text difficult to follow.  

In this part, they should also explain how PD symptoms can be measured and validated in zebrafish.

Why did the authors create a separate section focusing only on paraquat, but not on other chemicals mentioned?

The last sentence in the first part “The convergence of oxidative stress, neuroinflammation, and mitochondrial dysfunction thus explains the pathological mechanism underlying PQ-induced PD in zebrafish” should be expanded with a more detailed description of the symptoms, as the same description could also apply to the pathomechanism of Alzheimer’s disease.

The table is not very informative and should be modified to provide more detail, for example, precise results obtained using such models.

To summarise: In my opinion, the manuscript requires considerable modifications to achieve better quality.

Comments on the Quality of English Language

Through the text, authors using both British and American spelling, whereas should be consistent to one. I don't have any further comments about quality of language.

Author Response

Comment 1: The manuscript is too general; it lacks specific zebrafish PD research examples.
Response: We thank the reviewer. We have now included several specific research examples covering toxin exposure (MPTP, rotenone, paraquat) and genetic models, highlighting behavioral changes, molecular insights, and pathological outcomes. These updates are highlighted in yellow in the revised manuscript.

Comment 2: There is repetition of PD symptoms across paragraphs.
Response: We appreciate this observation. The manuscript has been carefully revised to remove redundancy. Repeated descriptions of PD symptoms have been consolidated, and where retention was necessary, additional insights were added. The edits are highlighted in yellow.

Comment 3: Why only the paraquat section?

Response: We acknowledge this important question. We created a dedicated section on paraquat because our laboratory has conducted original experiments using paraquat to establish a zebrafish PD model. This provided us with unique insights into paraquat’s mechanisms, symptoms, and validation as a disease model. For this reason, paraquat was treated in greater detail compared to other toxins.

Comment 4: The last sentence on the PQ section should be expanded to distinguish from Alzheimer’s disease.

Response: The text has been expanded to provide a detailed comparison of overlapping and distinct symptoms and mechanisms between PD and Alzheimer’s disease, including motor, cognitive, and neuropsychiatric features as well as oxidative stress, mitochondrial dysfunction, and neuroinflammation. These clarifications are highlighted in yellow.

Comment 5: The table is not very informative and should provide precise results.
Response: We thank the reviewer for this constructive feedback. As per the previous reviewer's comment, we included phenotypic features and a comparison of the zebrafish model with other rodent models and their advantages. Reflecting to the current reviewer comment, we did not update the table

Comment 6: Language consistency (British vs. American).
Response: The manuscript has been carefully checked for consistency. We have standardized to American English throughout.

Reviewer 3 Report (New Reviewer)

Comments and Suggestions for Authors

The work addresses a topic of great scientific relevance, not only regarding Parkinson's disease but also the use of the zebrafish model in research. The text is extremely objective, clear, and provides excellent observations on the state of the art in this area of ​​research.

Just a few observations:

In the abstract, the phrase "research due to their genetic manipulation" needs to be rewritten. It would be better if it were written "due to the possibility of genetic manipulation," which would leave less ambiguity.

Table 1 is not a table, but a chart, as the structure is closed on the sides.

Author Response

Comment 1: Abstract phrase “research due to their genetic manipulation” should be rewritten.
Response: Corrected as suggested to “due to the possibility of genetic manipulation.” The revised sentence is highlighted in yellow.

Comment 2: Table 1 is not a table but a chart.

Response: We thank the reviewer for this observation. As per the previous reviewer's comment, we included phenotypic features and a comparison of the zebrafish model with other rodent models and their advantages.

Round 2

Reviewer 2 Report (New Reviewer)

Comments and Suggestions for Authors

Authors, make all the suggested changes and explained the intriguing questions. 

I still have a remark, that I still can spot mix of UK and US English.

Overall, I don't have any more commentary to add.

This manuscript is a resubmission of an earlier submission. The following is a list of the peer review reports and author responses from that submission.

Round 1

Reviewer 1 Report

Comments and Suggestions for Authors

The review is not comprehensive and does not add to the growing body of literature regarding the use of zebrafish for PD research. 

Author Response

Point-by-Point Response to Reviewers

Manuscript ID: biology-3625291

Manuscript Title: Tiny Fish, Big Hope: Zebrafish Unlocking Secrets to Fight Parkinson’s Disease

To the Editor and Reviewers:

We sincerely thank the editor and reviewers for their insightful feedback on our manuscript. We have carefully revised the manuscript in response to the comments and believe the quality and clarity of the work have improved significantly as a result. Below is our detailed, point-by-point response.

Reviewer 1

Comment: The review is not comprehensive and does not add to the growing body of literature regarding the use of zebrafish for PD research.

Response: We thank the reviewer for this feedback. We have now added several recent and relevant original studies (e.g., Mazzolini et al., 2020; Wang et al., 2018; Prabhudesai et al., 2016) that improve the comprehensiveness of our literature review. In addition, we have added a comparison table (Table 1) to provide clarity for readers less familiar with model organism differences.

Reviewer 2 Report

Comments and Suggestions for Authors

The review article by Manjunath et al., entitled Tiny Fish, Big Hope: Zebrafish Unlocking Secrets to Fight Parkinson’s Disease, describes the use of zebrafish as a model organism for Parkinson's Disease, with focus on the study of degeneration, neuroinflammation, and mitochondrial pathology. The use of zebrafish in neuroscience generally and in PD research specifically, is constantly growing given the advantages that zebrafish offer in terms of genetic tools availability and high regenerative capacity.  This work supports their translational potential in neurodegenerative disease research.

However, there are some areas that be improved. Please see below:

  • Even though reviews can cite other reviews, it is generally preferred to cite original research papers instead.
  • There are several grammatical inconsistencies and informal phrasing, such as “PD researches,” that disrupt the scientific flow.
  • The advantage of zebrafish being transparent is repeated throughout the text in different sections. If repetition is essential and cannot be avoided, please add some new insights to avoid redundancy.
  • The discussion of paraquat and MPTP, can be strengthened by more explicitly mentioning the specific advantages and limitations of each.
  • Consider adding a schematic figure or summary table outlining key phenotypes, modelling approaches, and comparative advantages relative to rodent systems. They provide useful reference points for readers, especially those less familiar with zebrafish as a model.
  • The future directions section appears as general and hypothetical. More specific suggestions can be included, that can help improve translation potential to human biology.
Comments on the Quality of English Language

The manuscript presents valuable scientific content and is generally readable; however, the quality of the English language requires attention. There are frequent grammatical errors, awkward phrasing, and informal constructions that occasionally obscure meaning and detract from the professional tone of the manuscript. Addressing these language issues will significantly enhance the clarity, professionalism, and accessibility of the manuscript, allowing the scientific contributions to be more effectively communicated to the broader research community.

Author Response

Point-by-Point Response to Reviewers

Manuscript ID: biology-3625291

Manuscript Title: Tiny Fish, Big Hope: Zebrafish Unlocking Secrets to Fight Parkinson’s Disease

To the Editor and Reviewers:

We sincerely thank the editor and reviewers for their insightful feedback on our manuscript. We have carefully revised the manuscript in response to the comments and believe the quality and clarity of the work have improved significantly as a result. Below is our detailed, point-by-point response.

Reviewer 2

  1. Prefer original research papers over reviews.
    Response: We agree. We reviewed the citations and replaced or supplemented reviews with original studies where appropriate.

    2. Grammatical inconsistencies and informal phrases (e.g., “PD researches”).
    Response: We thoroughly edited the manuscript for grammar, sentence structure, and scientific tone.

    3. Repeated mention of zebrafish transparency.
    Response: Repetition has been addressed. The discussion of transparency now appears in a single, focused section.

    4. Strengthen paraquat and MPTP discussion with clear advantages/limitations.
    Response: We revised that section to include a direct comparison of the specific mechanisms, benefits, and drawbacks.

    5. Add a schematic or summary table comparing zebrafish with rodent PD models.
    Response: We created and included Table 1 in the manuscript.

    6. Make 'Future Directions' section more specific.
    Response: The section has been revised to include actionable ideas like optogenetics and behavioral tracking.

Reviewer 3 Report

Comments and Suggestions for Authors

In the paragraph entitled "Zebrafish as a model organism to study Parkinson’s disease", I suggest to the authors to critically review the paragraph  based on the recent literature published about the animal model and Parkinson disease expecially about genes and proteins involved in the disease. 

Author Response

Point-by-Point Response to Reviewers

Manuscript ID: biology-3625291

Manuscript Title: Tiny Fish, Big Hope: Zebrafish Unlocking Secrets to Fight Parkinson’s Disease

To the Editor and Reviewers:

We sincerely thank the editor and reviewers for their insightful feedback on our manuscript. We have carefully revised the manuscript in response to the comments and believe the quality and clarity of the work have improved significantly as a result. Below is our detailed, point-by-point response.

Reviewer 3

Comment: Critically review the section "Zebrafish as a model organism to study Parkinson’s disease" using recent literature.
Response: We revised the section to include recent studies and references on α-synuclein, Parkin, and LRRK2.

Language Quality

Comment: The manuscript has grammatical errors and awkward phrasing.
Response: The manuscript has undergone full language editing to improve clarity and professionalism.

We hope these changes meet your expectations and sincerely thank you for your efforts in improving our work.

Round 2

Reviewer 1 Report

Comments and Suggestions for Authors

The authors added only 4 references to primary literature without . This is still not acceptable for a literature review. 

Author Response

We sincerely thank the reviewer for this valuable comment. In the revised manuscript, we have substantially expanded the number of primary literature references to strengthen the scientific foundation of the review. Specifically, we added, ensuring that the manuscript now contains a balanced and comprehensive coverage of original research articles alongside review papers.

All added references are highly relevant to the scope of the manuscript and were critically selected to enhance the depth and quality of the discussion. To facilitate the review process, all newly added references have been highlighted in the revised manuscript.

Additionally, we have ensured that the reference formatting follows the journal template:

  • If >10 authors, “et al.” is used.
  • If ≤10 authors, all author names are listed.